# The Effect of Physiotherapy Group Intervention in Reducing Pain Disorders and Mental Health Symptoms among Syrian Refugees: A Randomized Controlled Trial

**DOI:** 10.3390/ijerph17249468

**Published:** 2020-12-17

**Authors:** Wegdan Hasha, Jannicke Igland, Lars T. Fadnes, Bernadette Kumar, Jasmin Haj-Younes, Elisabeth Marie Strømme, Eirin Zerwekh Norstein, Rolf Vårdal, Esperanza Diaz

**Affiliations:** 1Department of Global Public Health and Primary Care, University of Bergen, Årstadveien 17, 5020 Bergen, Norway; Jannicke.Igland@uib.no (J.I.); lars.fadnes@uib.no (L.T.F.); Jasmin.Haj-Younes@uib.no (J.H.-Y.); Elisabeth.Stromme@uib.no (E.M.S.); Esperanza.Diaz@uib.no (E.D.); 2Department of Addiction Medicine, Haukeland University Hospital, Jonas Lies vei 65, 5021 Bergen, Norway; 3Norwegian Institute of Public Health, Unit for Migration and Health, 222 Skøyen, 0213 Oslo, Norway; BernadetteNirmal.Kumar@fhi.no; 4OsloMet—Faculty of Health Sciences, Oslo Metropolitan University, P.O. Box 4, St. Olavsplass N, 0130 Oslo, Norway; eirin.zn@gmail.com; 5Center for Migration Health, Solheimsgaten 9, 5058 Bergen, Norway; Rolf.Vardal@bergen.kommune.no

**Keywords:** group intervention, pain, mental health, Syrian refugees, randomized controlled trial

## Abstract

Chronic pain is common among refugees, and often related to mental health problems. Its management, however, is often challenging. A randomized waitlist-controlled trial was designed to study the effect of group physiotherapy activity and awareness intervention (PAAI) on reducing pain disorders, and secondarily improving mental health, among Syrian refugees. A total of 101 adult Syrian refugees suffering from chronic pain were randomized to either the intervention group or the control group, which thereafter also received PAAI after a waiting period. Pain intensity measured by the Brief Pain Inventory (BPI) was the primary outcome. Scores from the Impact of Events Scale-Revised (IES-R 22) and the General Health Questionnaire (GHQ-12) were secondary outcomes. Intention-to-treat analyses (ITT) showed no effect of the intervention on either pain levels (regression coefficient [B {95% CI} of 0.03 {−0.91, 0.96}], IESR scores [4.8 {−3.7, 13.4}] or GHQ-12 scores [−0.4 {−3.1, 2.3}]). Yet, participants highly appreciated the intervention. Despite the negative findings, our study contributes to the evidence base necessary to plan targeted and effective health care services for refugees suffering from chronic pain and highlights the challenge of evaluating complex interventions adapted to a specific group.

## 1. Introduction

Armed conflicts and war often result in displacement and mass migrations. More than one million Syrians reached the European Union in 2015 [1], becoming the largest refugee group in many European countries, including Norway [2,3].

Refugees are at high risk of developing chronic pain [4,5,6]. The reported prevalence of chronic pain among Syrian refugees varies between 30% in Norway [7] and 37% in Australia [8]. Physical limitations, such as walking difficulties, are described for 47% of Syrian refugees in Lebanon [9]. However, chronic pain is understudied in this group, and the literature often focuses primarily on mental health, especially post-traumatic stress disorder (PTSD) [10]. Indeed, poor mental health and chronic pain reinforce each other [11]. Chronic pain can disturb daily functioning, which can lead to mental distress [12,13], and mental health problems can have a negative impact on the experience of pain [14].

In 2015, a study from Norway reported that the prevalence of chronic pain among those attending an outpatient mental health clinic was 66% in the host population versus 88% among traumatized refugees [4]. Similar findings are reported from Denmark, the United Kingdom, and Germany [14,15,16]. In other comorbidity studies, the prevalence of PTSD among persons with chronic pain ranges from 10% to 50%, while the prevalence of chronic pain among persons with PTSD ranges from 20% to 80% [17,18].

The understanding and management of chronic pain is influenced by tradition and culture [19]. When arriving in the host country, refugees should be offered appropriate and culturally acceptable forms of health care. In the Nordic countries, combinations of basic body awareness therapy (BBAT) and general physiotherapy exercises integrating Eastern and Western traditions have been used as treatment [20,21,22]. BBAT is reported to have a positive impact among patients with chronic pain and mental health symptoms [20,23,24]. Psychomotor physiotherapy was developed in Norway and focuses on increasing awareness of the body–mind relationship [25]. However, there is little evidence of the effect of physiotherapy on refugees in either pain or mental health symptoms.

While individual treatment allows for personalized therapy, group dynamics might be preferred by refugees, who often lack social contact with peers upon arrival in the new country [26,27]. A recent systematic review and meta-analysis found that group and individual physiotherapy delivered similar results and that group programs may need to be considered more often [28]. However, there is a lack of evidence of the effect of group physiotherapy versus individual physical activity, as part of the treatment for chronic pain among refugees [29,30].

The University of Bergen (Norway), in collaboration with the Center for Migration Health at the municipality of Bergen, developed a new group physiotherapy activity and awareness intervention (PAAI) based on previous clinical experience with group interventions for refugees [31] and influenced by principles found in BBAT and psychomotor physiotherapy. We hypothesize that PAAI conducted in groups would ineffectively help reduce pain disorders among Syrian refugees and also improve mental health. The aim of this study was to evaluate this effect, as, and also participants’ experiences of the intervention.

## 2. Materials and Methods

### 2.1. Study Design

This RCT is part of the main study ‘Changing health and Health care Needs Along the Syrian refugee’s Trajectories to Norway ‘(CHART). The trial was registered with Clinical Trials.gov on https://clinicaltrials.gov/ct2/show/NCT03951909. To include user participation in the design of the interventions, the study was retrospectively registered on 19 February 2019. The regional ethics committee approved the study in June 2018 (REK 2018/603), and informed written consent was given by all the participants.

The protocol of the intervention studies in CHART has been previously described following the SPIRIT recommendations [31]. CHART includes two randomized waitlist-controlled trials, one testing a self-help group intervention for mental health symptoms, not described in this paper, and the other, reported here, testing a physiotherapy activity and awareness intervention. Given the prevalent coexistence of chronic pain and mental health problems among refugees, participants with pain symptoms and/or mental health problems were recruited between 2018 and 2019 for both trials at the same time. As described in the protocol, participants with the greatest burden of pain symptoms were allocated to the PAAI intervention and randomized to either an immediate intervention group or a delayed intervention group, which received the same intervention but at a later point in time for ethical reasons. For the rest of the paper, we refer to the groups as intervention and control despite the fact that the last group also received the intervention after the evaluation of the main results. Details regarding sample size calculations have previously been described (31). An initial sample size of 34 participants in each group was calculated to be able to detect an intervention effect of 3 points on the BPI pain intensity scale participants with 80% power and a significant level of 5% and assuming 25% attrition.

### 2.2. Participants

As reported in the protocol [31], Syrian refugees 16 years or older living in Bergen and adjacent municipalities, with pain symptoms and/or mental health problems, were informed about the study in Arabic by the first author at different arenas. Those who consented to participate filled out a baseline questionnaire (Q0) in Arabic, under the guidance of a bilingual field worker. Among the 180 recruited, 101 had predominant burden of pain symptoms compared to mental health symptoms and were included in the PAAI trial reported in this paper. Of these, 50 participants were randomized to the intervention group and 51 to the control group. A total of 29 participants withdrew after randomization before initiation of the interventions. Thirty-eight participants (76%) from the intervention group and 34 (67%) from the control group attended the first session and completed the questionnaire the day the group sessions began (Q1a). Thirty-one participants (62%) from the intervention group and 24 (47%) from the control group completed the same questionnaire (Q1b) eight weeks after the group sessions ended. Twenty-seven participants (54%) from the intervention group and 23 (47%) from the control groups answered the questionnaire (Q1c) 12 weeks after the group sessions ended. Figure 1 shows the CONSORT flow chart. The number of participants who attended the PAAI sessions in both groups are summarized, by gender, in Appendix A.

### 2.3. Inclusion and Exclusion Criteria

Participants who reported pain symptoms that lasted over six months and scored 3 or higher in any of the average or current pain severity scores of the BPI were eligible for inclusion. Participants who lived far away from the therapy locations were excluded from the study. Those who scored 25 or higher on the GHQ-12 or 37 or higher on the IES-R 22 were assessed by a psychologist prior to their involvement in the study to decide the suitability of group treatment, since high scores could indicate serious mental health problems. The requirement of close medical follow-up (e.g., for those receiving treatment for diabetes with complications or for cancer) was also considered as exclusionary criterion, although no participant was excluded for these reasons.

### 2.4. Intervention

As explained in detail in the protocol, eight Syrians of both genders and not related to the study were consulted, on the practical implementation of PAAI, specifically with regard to the gender issues and session timing [31]. Physiotherapists with previous experience working with refugees led the PAAI sessions weekly for 8 weeks in groups of up to 10 persons. Each session lasted an hour and included the same key elements each time: mindfulness exercises and ball games warm-ups, followed by exercises involving sitting on a chair, laying down, relaxation, coordination and breathing exercises, and a short closing round [31]. The sessions were held in Norwegian with an Arabic interpreter, separately for women and men, and by a therapist and an interpreter of the same gender as the group. Instructions, phrases, and particular words used by therapists were shared with interpreters prior to the sessions. Participants were advised to pay attention to their own limitation regarding pain and range of motion, and were actively invited to comment on their own feelings during or after the sessions and report to the physiotherapists any injury or worsening of pain related to the sessions.

In order to assess whether the intervention met with the initial plan and to monitor activity in the groups, the first author conducted systematic observation, taking detailed notes following a predesigned protocol (Appendix B, Figure A1) during the sessions, at least twice in each group. The therapists kept the agreed plan as described in the protocol. The only difference noticed in the dynamic of the groups was related to gender. In the female groups, all were active, curious, interested, expressed satisfaction, acted friendly towards each other, and seemed to have fun. The male groups were less active, with some participants seeming less satisfied and not networking as much with each other. The interpreter’s Arabic dialect was not necessarily Syrian on every occasion, but there were no signs of misunderstandings.

### 2.5. Outcomes

The primary outcome was the degree of pain measured by the pain intensity domain of BPI, calculated as the mean of the four items on pain intensity (range 1–10). The secondary outcomes were scores from the IES-R 22 and GHQ-12 questionnaires [31]. All outcomes were measured right after the intervention in the intervention group and right before the start of the delayed intervention in the control group. The same outcomes were used for the additional longitudinal analyses of change during the intervention phase for both groups combined.

### 2.6. Measures

Two similar questionnaires, Q0 and Q1, were translated into Arabic and developed, as explained in the protocol [31]. Both included questions on socio-demographics and migration, and health status, including chronic pain and mental health. Q0 was longer and used to identify participants at baseline. Q1 was shorter and was applied three times: at the beginning of the intervention (Q1a), after eight weeks, i.e., at the end of the intervention (Q1b), and 12 weeks after the first session of the intervention, i.e., four weeks after the intervention ended (Q1c).

Pain in the last six months was measured using the four pain severity items (worst pain, least pain, average pain, pain right now) included in the BPI—Short Form (BPI-SF). Each of the items can be scored from 0 to 10 [31]. The IES-R 22 and the GHQ-12 were used to assess mental health. IES-R 22 measures the subjective response to a specific traumatic event in adult populations, especially with respect to intrusion, avoidance, and hyper-arousal. The three areas together provide a total subjective stress IES-R score (range 0–88). IES-R is not a diagnostic tool. The GHQ-12 scale is developed for the general population, as opposed to patients, asking whether the respondent has experienced a particular symptom or behavior recently. It includes 12 items, each of which is rated on a 4-point scale (less than usual, no more than usual, rather more than usual, or much more than usual), giving a maximum total score of 36 (range 0–36). Both instruments have been validated among Arabic-speaking refugees [32,33].

Additionally, to gain an insight into experiences with the intervention, we conducted a simple qualitative explorative study with a phenomenological design. After the group interventions, individual semi-structured interviews were conducted in Arabic by the first author (Figure A2). Questions in these interviews where adapted from Sarkadi et al., in their evaluation of a group intervention for unaccompanied refugee minors [34]. In total, 17 interviews were conducted, 11 with participants who had fulfilled the intervention and six with participants who dropped out before or during the sessions, from both the intervention and the control (delayed intervention) groups.

### 2.7. Randomization and Blinding

A block randomization was conducted using a 1:1 allocation ratio, with block sizes of 4, 6, and 8 generated by a statistician using the rollac command in Stata software, version 15. To provide interventions in suitable time, recruitment was planned in three waves, followed by three rounds of interventions. The time difference in starting the intervention between the intervention and control groups was eight weeks. The intervention was not blinded, neither for the participants, nor for the instructors or authors. The first author recruited the participants and assessed outcomes but did not have access to the randomization list.

### 2.8. Statistical Analysis

Baseline characteristics are presented for the intervention and control groups by means and standard deviations for continuous variables and counts and percentages for categorical variables. The intention-to-treat (ITT) principle was used. The effect of the intervention on the primary outcome was assessed using linear mixed models with random intercept for individuals and the BPI pain intensity score as a dependent variable (Equation (1)). Data were analyzed in long format (two observations per individual), with the BPI score at Q0 as the first BPI measurement for each individual. The second measurement was defined as the measurement at the last treatment session (eight weeks) in the intervention group and the measurement at the first session for the control group. The following constrained linear mixed model was estimated:BPI_i_ = β_0 + β_1∙TIME + β_2∙TIME∙GROUP + b_0_i_ + ε(1)
where TIME is a binary variable (0/1) with Q0 (time of randomization) as the reference, GROUP is a binary group allocation variable with the control group as the reference, and b0_i_ is the random intercept for each individual i. The intervention effect was estimated as the interaction effect between GROUP and TIME. By omitting the main effect of group from the model, we achieved an adjustment for baseline differences in BPI [35]. Corresponding models were estimated for the secondary outcomes IES-R and GHQ and also for BPI pain interference as a supplementary off-protocol outcome. The intervention effect was reported as regression coefficients for the interaction term with 95% confidence intervals and can be interpreted as the mean difference in change in outcome score after eight weeks between the intervention and control groups after adjustment for potential differences in outcome at baseline. In line with the ITT principle, we included all 101 individuals who were randomized in the model. All individuals had baseline measurements (Q0), but in total, 36 participants were lost to follow-up and had missing values on the second measurement. The approach with linear mixed model data in long format provides unbiased estimates of the intervention effect as long as follow-up data are missing at random, given the covariates included in the model. As a sensitivity analysis, we repeated the linear mixed models with adjustment for variables that were significantly different between participants with complete data and participants who were lost to follow-up. As a second sensitivity analysis, we also applied mixed effects linear regression with random intercept and slope for group membership to investigate if differences in group dynamics within each recruiting wave could influence the results.

In longitudinal analyses of changes during the intervention phase and the four weeks after the last session for the intervention and control (delayed intervention) groups combined, we used data in long format with three observations per person and applied linear mixed effects regression with random intercept for each individual. Inclusion of the random slope for time did not improve model fit and we therefore only estimated the fixed effect for time. Time was modelled both as a categorical covariate with the first session as the reference and as a continuous covariate with the values 0, 8, and 12 in order to test for a linear trend over weeks. Differences in change over time between genders were investigated by stratification and by inclusion of an interaction-term between time and gender.

Stata SE version 16 was used to analyze the data. All tests were two-sided, with a 5% level of significance.

### 2.9. Qualitative Analysis

The interviews were recorded and transcribed. Two Arabic speakers translated the interviews from Arabic to English, which were then sent to a professional translator with the original records for quality assurance. A master student (EZN) coded and analyzed the interviews in close collaboration with supervisors. No specific analysis software was used. Systematic text condensation was then used in the analysis, which consisted of four steps: (1) achieving a total impression—from chaos to themes; (2) identifying and sorting meaning units—from themes to codes; (3) condensation—from code to meaning; and (4) synthesizing—from condensation to descriptions and concepts, as described by Malterud [36,37]. Only the main themes are presented here for the sake of parsimony.

## 3. Results

The baseline characteristics of the participants randomized to intervention and control groups (*n* = 101) are presented in Table 1. Participants in both groups were similar in most respects but were younger in the control group, and a higher percentage was married and had children in the intervention group. We found no differences in either BPI scores, IES-R 22 or GHQ-12, between the two groups at baseline. The groups were also balanced in terms of exposure to stressful events, self-reported health, and daily use of medication. From the 101 randomized participants, 65 had follow-up data on the main and secondary outcomes (31 in the intervention and 34 in the control groups). A comparison of baseline characteristics between the 65 participants with 2 measurements and the 36 participants with only baseline measurement is reported in Appendix A. The lost-to-follow-up less had stayed in a transit country before migrating to Norway, were more physically active, and had higher education but were otherwise similar to the actual participants.

Table 2 shows the ITT effect of the intervention on chronic pain (BPI) and mental health scores (IES-R 22 and GHQ-12), based on the scores at baseline and after treatment for the intervention group and at baseline and at the end of the waiting period for the control group. Intra-cluster correlation coefficients for correlation of repeated measures within individuals were 0.66, 0.61, and 0.68 for BPI, IES-R 22, and GHQ-12, respectively. The ITT analyses comparing the intervention against the control groups after adjustment for baseline BPI measurement showed no effect of the intervention on pain levels, with a regression coefficient (95% CI) of 0.03 (−0.91, 0.96). With regard to mental health, IESR scores increased non-significantly in the intervention group by 4.8 points, and did not change in the control group, and there was no significant effect of the intervention. GHQ-12 scores were reduced by 1.4 and 0.8 points in the intervention and control groups, respectively, but the intervention effect was not significant, with a regression coefficient (95% CI) of −0.4 (−3.1, 2.3). In the supplementary analysis of the pain interference inventory of the BPI, change in both the intervention and control groups was not significant, with regression coefficients (95% CI) of −0.67 (−1.52, 0.18) and 0.40 (−0.21, 1.01), respectively. The effect of the intervention was in the direction of increased pain interference but was not significant, with a regression coefficient (95% CI) of 0.97 (−0.17, 2.12).

As additional analyses, Figure 2 shows the longitudinal change in the mean levels of BPI, IES-R 22, and GHQ-12 separately for men and women and for all participants together (*n* = 101) taking into account the intervention period for the control group, after having waited for the treatment. For BPI and GHQ-12, there was a significant reduction from the first session (week 0) until the last session (week 8) for both men and women, and the measurement in week 12 was also significantly reduced compared to week 0, as summarized in Appendix A. The test for linear change with week treated as a continuous covariate was significant in both men and women for BPI and GHQ-12. The reduction was somewhat larger in women compared to men, but the test for interaction between gender and week was not significant (*p* = 0.69 for BPI and *p* = 0.85 for GHQ). For IES-R, there were no significant changes over time in either men or women.

### Participants’ Reflections on the Intervention

The qualitative analyses revealed three main themes. First, the participants greatly appreciated the sessions as a meeting place or social support platform for their community to gather and talk about the common challenges of being newcomers to Norway. Second, the intervention equipped the participants with tools to apply in everyday life, like learning to focus on the body, which in turn led to better sleep, healthy seating, and including particular movements in their daily routine. These elements enhanced self-confidence and gave the participants a feeling of control over their own health. Third, the participants described the facilitators (e.g., good therapist, good interpreter, positive environment, the possibility of breaking everyday routines, etc.) and barriers (e.g., session timings that clashed with family duties or medical appointments, especially for women, or dissatisfied gossipers) to their participation in the intervention.

## 4. Discussion

Our study focuses on chronic pain among refugees, adding new important information to the research that has previously focused mostly on mental health problems. To the best of our knowledge, this is the first trial to study the effects of physiotherapy group intervention among refugees with chronic pain. From an ITT point of view, our physiotherapy group intervention (PAAI) had no effect on reducing either pain disorders or improving mental health in our sample of refugees. However, the participants were satisfied with the intervention, and reported that it had provided them with the tools to apply in everyday life.

When comparing our study with previous research, we found only one pragmatic trial recently conducted among traumatized refugees with PTSD in Denmark in which participants were randomized to receive either individual BBAT or two other types of individualized therapy. This study showed no significant difference in improvement between groups in the ITT analysis, but a small significant improvement in the longitudinal analysis [38], which is similar to our study. This might be explained because PAAI did not specifically target either trauma or mental health. While other interventions directly targeting traumatic experiences like eye movement desensitization and reprocessing (EMDR) and emotional freedom techniques (EFTs) are effective for the treatment of trauma [39], they might not help with chronic pain, which was our main outcome. Our trial found no clear overall effects of the intervention itself in pain reduction after 8 and 12 weeks. There are several potential explanations for this. It is possible that the intervention was not sufficiently adapted for the participants and their situations. We found that 52% of the intervention group, and 45% of the control group said they never exercised. This is a difficult barrier to overcome and might necessitate a different kind of intervention. A third of the participants did not even attend the first meeting, and the qualitative interviews indicate that participating in the intervention could to some degree come into conflict with other activities and could contribute some degree of stress to an already busy life. Compared to the lost-to-follow-up, the participants had a longer migrating path to Norway, were less physically active, and had lower education, and it is also possible that the participant selection process involved a high proportion of treatment-resistant participants who might have tried and failed different treatments and were therefore difficult to help. Although we used validated tools to measure pain, pain can fluctuate, and we might have selected participants at a time when they had a very high degree of symptoms. In that case, any later measure will be more likely to be lower, in what is known as regression to the mean.

Among the Syrian refugees participating in this RCT, 68% were exposed to stressful events. All presented pain, it was an inclusion criterion, and most reported some degree of mental health symptoms. However, only 12% reported high levels of mental health problems. Previous RCT studies using physiotherapy among refugees have mainly focused on the effect it has on mental health symptoms like PTSD, major depression, anxiety, and stress [23,40,41,42,43]. These studies showed little effect of the treatment on these mental disorders. In the longitudinal analyses, our results seem to point to a differential effect of PAAI on mental health depending on the questionnaire used. While there was no effect or even worsening in scores for IES-R, the results were more positive for GHQ-12, used for the general population and not specifically for patients with mental health problems. This might indicate that PAAI could be better suited to improving mental health among persons with milder mental health symptoms. However, we did not have the power to stratify the analyses according to symptoms. More research into this field should be conducted, as mild mental health symptoms are very prevalent among refugees.

In two previous studies, the effect of BBAT among general psychiatric patients in Sweden, and among traumatized refugees in Denmark did not differ by gender [21,43]. This is in line with our study where no gender differences were found in the longitudinal analyses despite the slight difference in dynamics observed in both female and male groups.

Even if our intervention did not have an effect, there was statistically significant improvement in both groups with time, which might be due to factors other than the intervention or because of the Hawthorne effect where participants have a desire to please researchers. However, the results of our semi-structured interviews show that the group sessions were positive for the participants in several ways, including social and instrumental. Especially interesting are the qualitative results pointing to improvement in sleep quality, reduction of stress, and increase in self-confidence, which might decrease pain levels [40]. This improvement was also seen in a Swedish study using basic body awareness intervention [40,44], and should be further studied.

Our study has some important strengths. First, our trial was successful in terms of balance between the intervention and control groups. We managed to include the number of participants necessary to be able to find an effect in a group that is considered difficult to reach, since we had counted on a 30% drop out of participants based on physiotherapists’ previous experience. Related to this, the Arabic background of resource persons in the project, coupled with meetings with interpreters to discuss vocabulary beforehand, reduced language barriers. Second, the complementary use of both qualitative and quantitative methods to understand the effect of the intervention gives us a deeper understanding necessary for the implementation of future interventions in the real world [45]. Additionally, the close monitoring by observation of the sessions enabled us to understand the differences in dynamics between groups, depending on gender, that could be further investigated by statistical analyses. Lastly, there was close and well-established cooperation between the different organizations and municipalities in Norway, which facilitated recruitment of participants and implementation of the intervention. In collaboration with the municipality of Bergen, we chose to only include refugees from Syria, the largest migrant group in Norway since 2015, as including a range of groups could have complicated language translation in the sessions. However, we believe our study could also be generalized for other refugee populations suffering from chronic pain.

Our study has also limitations, including the relatively short intervention period and the limited length of the sessions, lasting only one hour each, which may have been insufficient. The delayed intervention design, judged as the only ethically acceptable solution, also made it difficult to assess potentially delayed effects that might exhibit at a later stage but were not present immediately after the intervention. In addition, outcome measures in our study were self-reported, and neither the participants, the outcome assessors, nor those conducting data analysis were blinded. However, we did our best to quality check the process in all its parts. Even though we had a consultant group to help us adapt the intervention to the lives of the participants, mandatory educational activities for newly resettled Syrian refugees, and issues related to childcare were the main reasons for participants dropping out from the session. This had an impact on attendance at the PAAI sessions, which was relatively low, reducing the power in our analyses and eventually introducing a potential selection bias. A low attendance rate, even lower than ours, is common in this type of study [38,42]. To overcome these limitations, future studies should try to tailor interventions to refugees’ lives even more and should ensure longer intervention periods.

## 5. Conclusions

Our study shows no effect of PAAI on either chronic pain or mental health symptoms among Syrian refugees living in Norway. However, participants were satisfied with the intervention and reported other beneficial effects and no side effects.

Our study contributes to the evidence base necessary to plan targeted and effective health care services for a vulnerable group and at the same time highlights the challenge of evaluating complex interventions adapted to a specific group.

## Figures and Tables

**Figure 1 ijerph-17-09468-f001:**
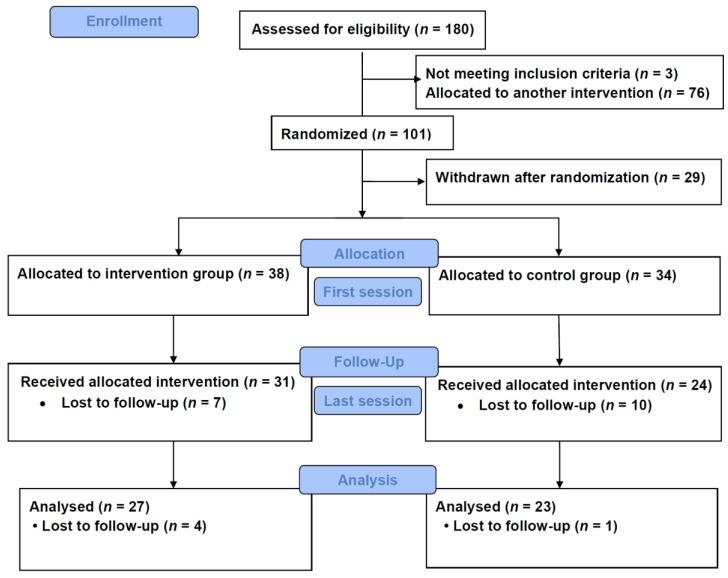
CONSORT flow chart.

**Figure 2 ijerph-17-09468-f002:**
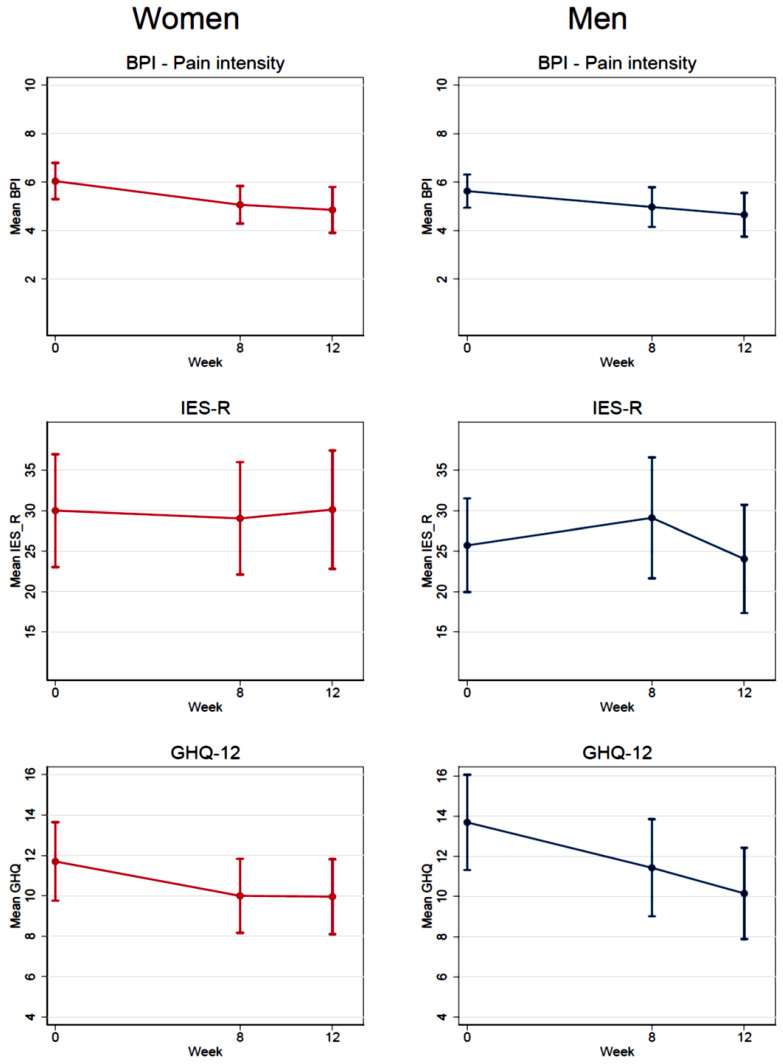
Levels of BPI (pain intensity), IES-R, and GHQ-12 for all participants at different intervals. Note: Week 0 = first PAAI session, week 8= last PAAI session and week 12 = four weeks after last session. Interventions and controls (after participating in the intervention -waitig list-) are presented together, stratified by gender. Mean scores with 95% confidence intervals.

**Table 1 ijerph-17-09468-t001:** Characteristics of the intervention and control groups at baseline.

		Intervention Group	Control Group
Total		50	51
Age (years), Mean (SD)		39 (11)	34 (11)
Low health literacy, *n* (%)		28 (56)	22 (43)
Female, *n* (%)		19 (38)	21 (41)
Ethnicity, *n* (%)	Arab	42 (84)	35 (69)
	Kurd	8 (16)	15 (29)
Stayed in any transit country on the way to Norway, *n* (%)		27 (54)	33 (65)
Marital status (married), *n* (%)		35 (70)	27 (53)
Have children, *n* (%)		39 (78)	29 (57)
Number of children, Mean (SD)		3 (1.4)	3 (1.4)
Education (years), Mean (SD)		9 (4.5)	9 (4.1)
Self-reported health, *n* (%)			
	Poor	21 (42)	20 (39)
	Neither	20 (40)	19 (37)
	Good	9 (18)	12 (24)

Self-reported diseases and daily use of medication, *n* (%)			
Physical or psychological pain at least 1 year		33 (66)	27 (53)
Physical pain more than6 months		36 (72)	39 (76)
Never do exercise		26 (52)	23 (45)
Rheumatic arthritis		9 (18)	10 (20)
Joint disease		33 (66)	38 (75)
Mental health problems		5 (10)	8 (16)
Headache		16 (32)	13 (25)
Daily use of painkillers		15 (30)	12 (24)
Daily use of psychotropics		3 (6)	7 (14)
Study outcomes			
IES-R, Mean (SD)	Intrusion (8–32)	9 (7.5)	9 (7.8)
	Avoidance (8–32)	9 (8.4)	10 (7.4)
	Hyper-arousal (6–24)	7 (5.8)	7 (5.9)
IES-R scores ≥37, *n* (%)		15 (30)	18 (35)
BPI scores	Having pain today (yes), *n* (%)	50 (100)	50 (98)
	Pain intensity (1–10), Mean (SD)	5.8 (1.9)	5.8 (2.3)
	Pain interference (1–10), Mean (SD)	4.4 (2.5)	4.5 (2.4)
GHQ-12 (0–36), Mean (SD)		12.7 (6.7)	12.5 (6.1)
GHQ-12 scores ≥25, *n* (%)		2 (4)	1 (2)

**Table 2 ijerph-17-09468-t002:** Effect of PAAI intervention on the primary and secondary outcomes. Intention to treat analyses using linear mixed models.

The Primary and Secondary Outcomes	Intervention, *n* = 50			Control, *n* = 51			Intervention Effect	
	Baseline (Q0). Mean (SD)	Last Session (Q1b). Mean (SD)	*p*-Value *	Baseline (Q0). Mean (SD)	End of Waiting Period (Q1a). Mean (SD)	*p*-Value *	B (95% CI)	*p*-Value
BPI	5.8 (1.9)	5.5 (2.1)	0.14	5.8 (2.3)	5.5 (2.1)	0.07	0.03 (−0.91, 0.96)	0.95
IES-R	25.9 (20.0)	31.2 (15.8)	0.10	26.7 (19.6)	25.8 (19.3)	0.90	4.8 (−3.7, 13.4)	0.27
GHQ-12	12.7 (6.7)	11.4 (6.1)	0.03	12.5 (6.1)	11.7 (5.3)	0.05	−0.4 (−3.1, 2.3)	0.76

Note: * Paired t-test for within-group change; PAAI = Physiotherapy activity and awareness intervention.

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
