# Peer review of "The Effect of Physiotherapy Group Intervention in Reducing Pain Disorders and Mental Health Symptoms among Syrian Refugees: A Randomized Controlled Trial"

_ijerph, 2020, doi:10.3390/ijerph17249468_

Round 1

Reviewer 1 Report

This manuscript describes a RCT of a novel group physiotherapy and awareness intervention trialed to reduce the burden of chronic pain specifically in a vulnerable patient population, namely Syrian refugees (n=101 included). While intention-to-treat analyses did not reveal any significant effects of the intervention of pain (measured via the BPI) or other health scales, the authors report that subjective/qualitative data supported the utility of the intervention, as a mechanism of peer support and empowerment for patients. Further, the authors discuss how these data add to the growing body of literature to better understand the issue of persistent pain within the refugee population.

The authors are to be commended on this original and interesting work, which provides important data on a marginalised patient population who are under-represented in the scientific literature. Strengths of this study include:

  • The authors use of active consumer engagement within their study protocol, to inform the design of the study intervention
  • The collection of complementary qualitative data (via interview) to provide stakeholder feedback on the intervention
  • The use of language translators and translated assessment tools, to engage a marginalised patient population
  • Attempts to monitor and report on the fidelity of the delivered intervention – something rarely done in research, but critical to understanding both the feasibility and effect of an intervention.

In order to strengthen the paper further, a number of minor issues have been outlined below for the authors consideration.

  • While the authors cite other publications, it would helpful to include more information about the content/format of the study intervention. (e.g. what was included in the sessions, what topics were covered, physical versus non-physical aspects of the program etc). At present it’s difficult to know exactly what the intervention entailed
  • The intervention in the present study appears to be single discipline therapy i.e. physiotherapy. Could you explain the rationale for using a single discipline – rather than multidisciplinary intervention (e.g. one that combined physiotherapy, psychological therapy and/or other modalities)? Given the growing consensus that holistic, multi-disciplinary treatment is most effective for the management of chronic pain, this may be a limited of the current intervention
  • The authors describe two related trials – one primarily targeting pain (reported here), and another primarily targeting mental health problems (reported elsewhere). Given the complex interplay between these conditions, could you explain your rationale for separating rather than combining these interventions? Did you measure mental health outcomes in the present study, and look for any effect of the intervention on mental health? (or conversely, did you measure pain outcomes in the mental health study?) This would be interesting to discuss
  • The authors included a single nationality of refugees in the present study (Syrian descent). Could they please comment on the reason for this choice, and whether this has implications for i) the content of the intervention, and ii) the generalisability of the study results?
  • The identified primary outcome was the intensity subscale of the BPI. Could the authors comment on why this particular measure (pain intensity) was chosen? In cases of chronic/persistent pain, measures of pain severity/intensity may not always be the most responsive to change. Often, even if pain persists or is still experienced, interventions like the one described help patients to better ‘live with’ and ‘manage’ their ongoing pain. Therefore, perhaps measures of pain interference, pain-related disability/distress and or pain self-efficacy may have been more informative, and may have been able to show change (such as the Pain Self Efficacy Questionnaire; the Pain Disability Index). Did the authors collect the interference subscale of the BPI? If so, this would be useful data to report.
  • Please clarify the methodology of block randomisation/group allocation – currently it is somewhat confusing. Also, why do participants appear to have been randomised before the consent process? (ie. in Figure 1, n=29 participants declined to participate after they were randomised)
  • Please confirm whether outcome assessors were blinded to group allocation – this is unclear. If not, this needs to be clearly stated as a study limitation and potential source of bias
  • Use of terminology – suggest using the term “waiting list control” or “randomised waitlist controlled trial” to describe the study design
  • Re Quali interviews – were these coded by 1 or 2 thematic coders? And was any software used for thematic analyses? (e.g. NVivo)
  • A power/sample size calculation has not been included. Could you please comment on required power for this study? Given that some results emerged as significant when the delayed data from the waitlist control group were pooled with the intervention group, this suggests that power may be an issue (and may be contributing to the negative findings). Important to note
  • Reporting of Linear Mixed Models requires additional values – would be useful to add T statistic, r value (for correlation between pre and post data), mean differences. Also, the explanation of LMM methodology in the methods (page 5), is somewhat grammatically confusing. Suggest grammatical editing
  • In Supplementary Table 1 – would be helpful to report percentages, as well as raw numbers (for ease of interpretation). Also, would be interesting to report the reasons for non-attendance at sessions where known

Overall, I think this is important and interesting work.

Author Response

Please, see attached file.

Reviewer 2 Report

Although the topic is very important due to its social application, I regret to inform to authors that the submitted manuscript does not fit with the journal standards.

The introduction section is very short and does not provide relevant information introducing the studied topic or the hypothesis.

The methods section does not follow the consort recommendation and there is a lack of information regarding the intervention and follow-up.

although it's plausible that in that situation, physiotherapy applications do not provide significant benefits, from my point of view the selected variables may not be the most appropriate.

I will encourage nevertheless to authors to promote this kind of intervention on these population groups for its valuable social role.

Author Response

Please, see attached file.

Reviewer 3 Report

This looks like it would be a great study however the introduction and the methods were not put in the manuscript. Instead, the directions of what the IJERPH expects in the sections was within the manuscript. 

This is unfortunate because the quantitative results and discussion of the study were well written. I could not understand everything (e.g. acronyms) because this is the type of information that would have previously described.

The qualitative results were not as thorough as there is  information missing that is usually accounted for in qualitative results. At minimum the demographics of the persons who participated, whether it was a  focus groups or individual interviews, a formal thematic analyses with a code book, codes, % of reporting the theme should be reported.

It is my expectation that the authors already have the sections that are missing and will be successful in a resubmission for a full review. 

Author Response

Please, see attached file.

Reviewer 4 Report

The submitted article appears carefully planned and executed. The conclusions are justified by the data and presented clearly. There are several comments this reviewer will make to improve what is already a well-written text:  

1. minor error in line 24 with "neither". Please change to "either" (correctly written in line 293).

2. The reader cannot tell what constitutes PAAI. Please include at least an abbreviated and operational definition so we know what was done. 

3. This is relevant given the authors' accurate and appropriate comments about how chronic physical pain is highly correlated with mental health issues, particularly trauma, sometimes reaching criteria for PTSD. There is no indication that trauma was targeted which would, for trauma-informed treatment providers, help to explain the lack of positive outcome.   4. Regarding the decision to use PAAI: The mainstream cognitive-behavior therapies are being shown to produce only modest results for chronic pain suffers whereas certain alternative treatments suggest robust results when targets involve treating pain as a somatic symptom of trauma. In Germany, for example, physician Jonas Tesarz has received awards for his combined scientific-clinical success with pain victims using EMDR, and EMDR is widely practiced in Norway and many other countries. Emotional freedom techniques (EFT) is also used around the world and while results are anecdotal, videotaped case examples are impressive (google Nick Orton). EMDR and EFT are both validated (in a study from Scotland) as effective treatment of trauma. Your reviewer suggests that these examples deserve further research as preliminary outcome findings suggest they may offer hope for chronic pain sufferers, including for those whose conditions are exacerbated by refugee status.

Thank you for your work.

Author Response

Please, see attached file.

Round 2

Reviewer 3 Report

I appreciate that the authors took the time to review the information provided by all and make the changes necessary. Because I did not see the original version in its entirety, I really counted on the reviews of those who did. And although I do wish there was more qualitative information, I respect the authors and their need to have this manuscript completed without doing an all out qualitative analysis of the work. I encourage you all to consider writing a separate paper that really dives into the themes that were found in the study. Overall I sincerely enjoyed reading this and commend them on a job well done!

Reviewer 4 Report

This reviewer appreciates the authors' willingness to consider the recommendations previously made. Best wishes on your future writings.